# GQA: Training Generalized Multi-Query Transformer Models from Multi-Head Checkpoints

**Joshua Ainslie,**[*] **James Lee-Thorp,**[*] **Michiel de Jong**[* †]
**Yury Zemlyanskiy**, **Federico Lebrón**, **Sumit Sanghai**

Google Research

## Abstract

Multi-query attention (MQA), which only uses a single key-value head, drastically speeds up decoder inference. However, MQA can lead to quality degradation, and moreover it may not be desirable to train a separate model just for faster inference. We (1) propose a recipe for uptraining existing multi-head language model checkpoints into models with MQA using 5% of original pre-training compute, and (2) introduce grouped-query attention (GQA), a generalization of multi-query attention which uses an intermediate (more than one, less than number of query heads) number of key-value heads. We show that uptrained GQA achieves quality close to multi-head attention with comparable speed to MQA.

## 1 Introduction

Autoregressive decoder inference is a severe bottleneck for Transformer models due to the memory bandwidth overhead from loading decoder weights and all attention keys and values at every decoding step (Shazeer, 2019; Pope et al., 2022; de Jong et al., 2022). The memory bandwidth from loading keys and values can be sharply reduced through *multi-query attention* (Shazeer, 2019), which uses multiple query heads but single key and value heads.

However, multi-query attention (MQA) can lead to quality degradation and training instability, and it may not be feasible to train separate models optimized for quality and inference. Moreover, while some language models already use multi-query attention, such as PaLM (Chowdhery et al., 2022), many do not, including publicly available language models such as T5 (Raffel et al., 2020) and LLaMA (Touvron et al., 2023).

This work contains two contributions for faster inference with large language models. First, we

show that language model checkpoints with multi-head attention (MHA) can be *uptrained* (Komatsuzaki et al., 2022) to use MQA with a small fraction of original training compute. This presents a cost-effective method to obtain fast multi-query as well as high-quality MHA checkpoints.

Second, we propose grouped-query attention (GQA), an interpolation between multi-head and multi-query attention with single key and value heads *per subgroup of query heads*. We show that uptrained GQA achieves quality close to multi-head attention while being almost as fast as multi-query attention.

## 2 Method

### 2.1 Uptraining

Generating a multi-query model from a multi-head model takes place in two steps: first, converting the checkpoint, and second, additional pre-training to allow the model to adapt to its new structure. Figure 1 shows the process for converting a multi-head checkpoint into a multi-query checkpoint. The projection matrices for key and value heads are mean pooled into single projection matrices, which we find works better than selecting a single key and value head or randomly initializing new key and value heads from scratch.

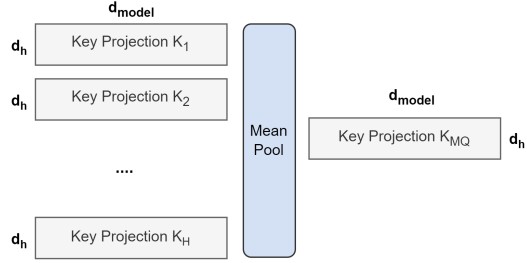

Figure 1: Overview of conversion from multi-head to multi-query attention. Key and value projection matrices from all heads are mean pooled into a single head.

The converted checkpoint is then pre-trained for

---

[*]Equal contribution.

[†]University of Southern California. Work done at Google Research.

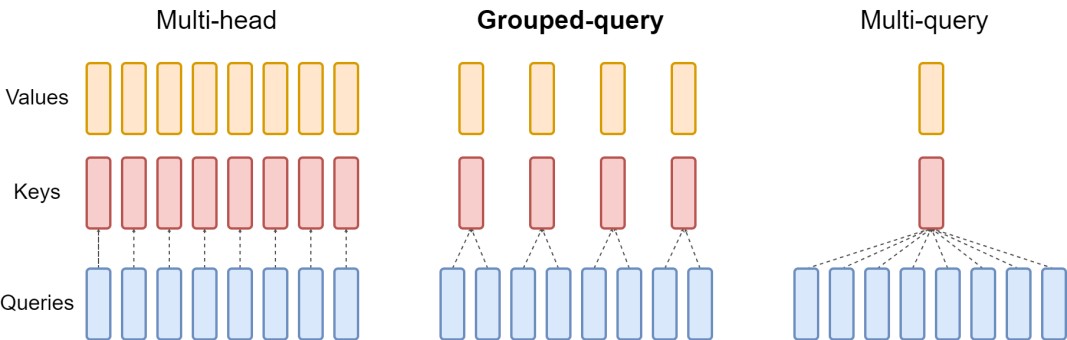

Figure 2: Overview of grouped-query method. Multi-head attention has H query, key, and value heads. Multi-query attention shares single key and value heads across all query heads. Grouped-query attention instead shares single key and value heads for each *group* of query heads, interpolating between multi-head and multi-query attention.

a small proportion $\alpha$ of its original training steps on the same pre-training recipe.

## 2.2 Grouped-query attention

Grouped-query attention divides query heads into *G groups*, each of which shares a single key head and value head. GQA-G refers to grouped-query with $G$ groups. GQA-1, with a single group and therefore single key and value head, is equivalent to MQA, while GQA-H, with groups equal to number of heads, is equivalent to MHA. Figure 2 shows a comparison of grouped-query attention and multi-head/multi-query attention. When converting a multi-head checkpoint to a GQA checkpoint, we construct each group key and value head by mean-pooling all the original heads within that group.

An intermediate number of groups leads to an interpolated model that is higher quality than MQA but faster than MHA, and, as we will show, represents a favorable trade-off. Going from MHA to MQA reduces $H$ key and value heads to a single key and value head, reducing the size of the key-value cache and therefore amount of data that needs to be loaded by a factor of $H$. However, larger models generally scale the number of heads, such that multi-query attention represents a more aggressive cut in both memory bandwidth and capacity. GQA lets us keep the same proportional decrease in bandwidth and capacity as model size increases.

Moreover, larger models suffer relatively less from memory bandwidth overhead from attention, as the KV-cache scales with model dimension while model FLOPs and parameters scale with the square of model dimension. Finally, standard sharding for large models replicates the single key and value head by the number of model partitions (Pope

et al., 2022); GQA removes the waste from such partitioning. Therefore, we expect GQA to present a particularly good trade-off for larger models.

We note that GQA is not applied to the encoder self-attention layers; encoder representations are computed in parallel, and memory bandwidth is therefore generally not the primary bottleneck.

## 3 Experiments

### 3.1 Experimental setup

**Configurations** All models are based on the T5.1.1 architecture (Raffel et al., 2020), implemented with JAX (Bradbury et al., 2018), Flax (Heek et al., 2020), and Flaxformer[1]. For our main experiments we consider T5 Large and XXL with multi-head attention, as well as uptrained versions of T5 XXL with multi-query and grouped-query attention. We use the Adafactor optimizer with the same hyperparameters and learning rate schedule as T5 (Raffel et al., 2020). We apply MQA and GQA to decoder self-attention and cross-attention, but not encoder self-attention.

**Uptraining** Uptrained models are initialized from public T5.1.1 checkpoints. The key and value heads are mean-pooled to the appropriate MQA or GQA structure, and then pre-trained for a further $\alpha$ proportion of original pre-training steps with the original pre-training setup and dataset from (Raffel et al., 2020). For $\alpha = 0.05$, training took approximately 600 TPUv3 chip-days.

**Data** We evaluate on summarization datasets CNN/Daily Mail (Nallapati et al., 2016), arXiv and PubMed (Cohan et al., 2018), MediaSum (Zhu et al., 2021), and Multi-News (Fabbri et al., 2019);

---

[1] https://github.com/google/flaxformer

| Model | $T_{infer}$ | Average | CNN | arXiv | PubMed | MediaSum | MultiNews | WMT | TriviaQA |
|-------|-------------|---------|-----|-------|--------|----------|-----------|-----|----------|
| | s | | $R_1$ | $R_1$ | $R_1$ | $R_1$ | $R_1$ | BLEU | F1 |
| MHA-Large | 0.37 | 46.0 | 42.9 | 44.6 | 46.2 | 35.5 | 46.6 | 27.7 | 78.2 |
| MHA-XXL | 1.51 | 47.2 | 43.8 | 45.6 | 47.5 | 36.4 | 46.9 | 28.4 | 81.9 |
| MQA-XXL | 0.24 | 46.6 | 43.0 | 45.0 | 46.9 | 36.1 | 46.5 | 28.5 | 81.3 |
| GQA-8-XXL | 0.28 | 47.1 | 43.5 | 45.4 | 47.7 | 36.3 | 47.2 | 28.4 | 81.6 |

Table 1: Inference time and average dev set performance comparison of T5 Large and XXL models with multi-head attention, and 5% uptrained T5-XXL models with multi-query and grouped-query attention on summarization datasets CNN/Daily Mail, arXiv, PubMed, MediaSum, and MultiNews, translation dataset WMT, and question-answering dataset TriviaQA.

translation dataset WMT 2014 English-to-German; and question answering dataset TriviaQA (Joshi et al., 2017). We do not evaluate on popular classification benchmarks such as GLUE (Wang et al., 2019) as autoregressive inference is less applicable for those tasks.

**Fine-tuning** For fine-tuning, we use a constant learning rate of 0.001, batch size 128, and dropout rate 0.1 for all tasks. CNN/Daily Mail and WMT use input length of 512 and output length 256. Other summarization datasets use input length 2048 and output length 512. Finally, TriviaQA uses input length 2048 and output length 32. We train until convergence and select the checkpoint with the highest dev performance. We use greedy decoding for inference.

**Timing** We report time per sample per TPUv4 chip, as measured by xprof (Google, 2020). For timing experiments we use 8 TPUs with the largest batch size that fits up to 32 per TPU, and parallelization optimized separately for each model.

## 3.2 Main results

Figure 3 shows average performance over all datasets as a function of average inference time for MHA T5-Large and T5-XXL, and uptrained MQA and GQA-8 XXL models with uptraining proportion $\alpha = 0.05$. We see that a larger uptrained MQA model provides a favorable trade-off relative to MHA models, with higher quality and faster inference than MHA-Large. Moreover, GQA achieves significant additional quality gains, achieving performance close to MHA-XXL with speed close to MQA. Table 1 contains full results for all datasets.

## 3.3 Ablations

This section presents experiments to investigate the effect of different modeling choices. We eval-

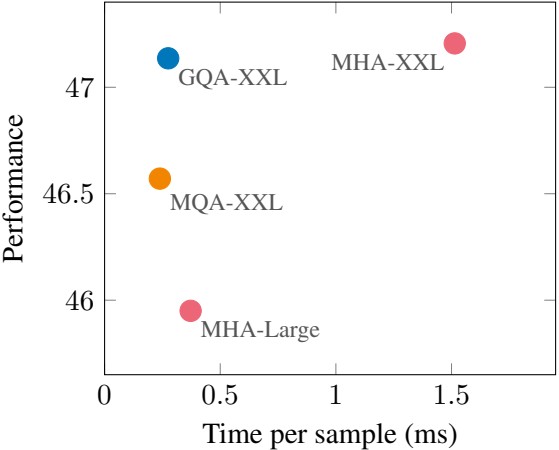

Figure 3: **Uptrained MQA yields a favorable trade-off compared to MHA with higher quality and faster speed than MHA-Large, and GQA achieves even better performance with similar speed gains and comparable quality to MHA-XXL.** Average performance on all tasks as a function of average inference time per sample for T5-Large and T5-XXL with multi-head attention, and 5% uptrained T5-XXL with MQA and GQA-8 attention.

uate performance on a representative subsample of tasks: CNN/Daily Mail, (short-form summarization), MultiNews (long-form summarization), and TriviaQA (question-answering).

**Checkpoint conversion** Figure 4 compares the performance of different methods for checkpoint conversion. Mean pooling appears to work best, followed by selecting a single head and then random initialization. Intuitively, results are ordered by the degree to which information is preserved from the pre-trained model.

**Uptraining steps** Figure 5 shows how performance varies with uptraining proportion for T5 XXL with MQA and GQA. First, we note that GQA already achieves reasonable performance after conversion while MQA requires uptraining to

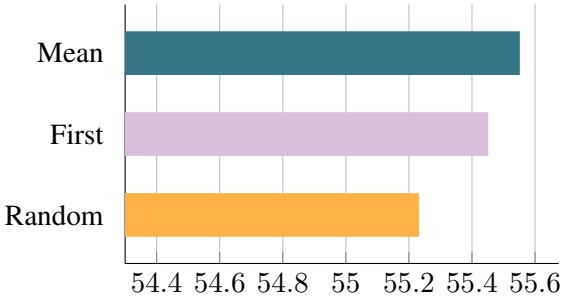

Figure 4: Performance comparison of different checkpoint conversion methods for T5-Large uptrained to MQA with proportion $\alpha = 0.05$. 'Mean' mean-pools key and value heads, 'First' selects the first head and 'Random' initializes heads from scratch.

be useful. Both MQA and GQA gain from 5% uptraining with diminishing returns from 10%.

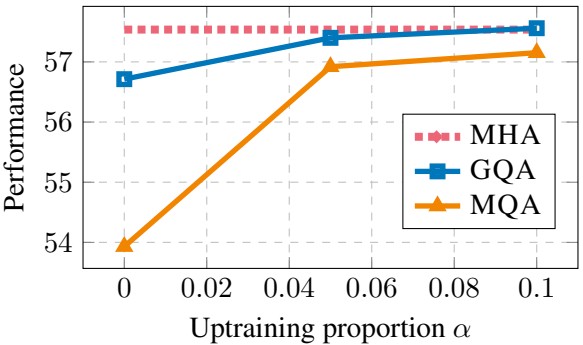

Figure 5: Performance as a function of uptraining proportion for T5 XXL models with MQA and GQA-8.

**Number of groups** Figure 6 demonstrates the effect of the number of GQA groups on inference speed. For larger models the memory bandwidth overhead from the KV cache is less constraining (Shazeer, 2019), while the reduction in key-value size is sharper due to the increased number of heads. As a result, increasing the number of groups from MQA only results in modest slowdowns initially, with increasing cost as we move closer to MHA. We selected 8 groups as a favorable middle ground.

## 4 Related Work

This work is focused on achieving a better trade-off between decoder quality and inference time through reducing the memory bandwidth overhead (Williams et al., 2009) from loading keys and values. Shazeer (2019) first proposed reducing this overhead through multi-query attention. Follow-up work showed that multi-query attention is espe-

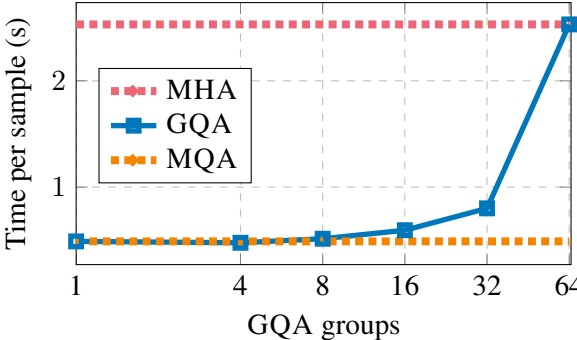

Figure 6: Time per sample for GQA-XXL as a function of the number of GQA groups with input length 2048 and output length 512. Going from 1 (MQA) to 8 groups adds modest inference overhead, with increasing cost to adding more groups.

cially helpful for long inputs (Pope et al., 2022; de Jong et al., 2022). Rabe (2023) independently developed GQA with public implementation.

A number of other methods have been proposed to reduce memory bandwidth overhead from keys and values, as well as parameters. Flash attention (Dao et al., 2022) structures the attention computation to avoid materializing the quadratic attention scores, reducing memory and speeding up training. Quantization (Dettmers et al., 2022; Frantar et al., 2022) reduces the size of weights and activations, including keys and values, by lowering precision. Model distillation (Hinton et al., 2015; Gou et al., 2021) instead reduces model size at a given precision, using data generated from the larger model to finetune the smaller model. Layer-sparse cross-attention (de Jong et al., 2022) eliminates most cross-attention layers which make up the primary expense for longer inputs. Speculative sampling (Chen et al., 2023; Leviathan et al., 2022) ameliorates the memory bandwidth bottleneck by proposing multiple tokens with a smaller model which are then scored in parallel by a larger model.

Finally, the uptraining procedure we propose is inspired by Komatsuzaki et al. (2022), which uptrains standard T5 checkpoints into sparsely activated Mixture-of-Experts models.

## 5 Conclusion

Language models are expensive for inference primarily due to the memory bandwidth overhead from loading keys and values. Multi-query attention reduces this overhead at the cost of decreased model capacity and quality. We propose to convert multi-head attention models to multi-query models

with a small fraction of original pre-training compute. Moreover, we introduce grouped-query attention, an interpolation of multi-query and multi-head attention that achieves quality close to multi-head at comparable speed to multi-query attention.

## Limitations

This paper focuses on ameliorating the memory bandwidth overhead from loading keys and values. This overhead is most important when generating longer sequences, for which quality is inherently difficult to evaluate. For summarization we employ Rouge score, which we know is a flawed evaluation that does not tell the whole story; for that reason, it is difficult to be certain our trade-offs are correct. Due to limited computation, we also do not compare our XXL GQA model to a comparitive model trained from scratch, so we do not know the relative performance of uptraining vs training from scratch. Finally, we evaluate the impact of uptraining and GQA only on encoder-decoder models. Recently, decoder-only models are extremely popular, and since these models do not have separate self-attention and cross-attention, we expect GQA to have a stronger advantage over MQA.

## Acknowlegements

We thank Santiago Ontañón, Afroz Mohiuddin, William Cohen and others at Google Research for insightful advice and discussion.

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

## A   Training Stability

We find that multi-query attention can lead to training instability during fine-tuning, in particular combined with long input tasks. We trained multiple T5-Large models with multi-query attention from scratch. In each case, pre-training suffered from frequent loss spikes and the final models diverged immediately when fine-tuning on long-input tasks. Uptrained multi-query attention models are more stable but still display high variance, so for multi-query models on unstable tasks we report average performance over three fine-tuning runs. Uptrained grouped-query attention models, however, appear to be stable, so we did not investigate futher on the root causes of multi-query instability.