# OpenReview forum: "GQA: Training Generalized Multi-Query Transformer Models from Multi-Head Checkpoints"
_EMNLP/2023/Conference — EMNLP 2023 Main_

### Official Review · Reviewer_B4s9 · 2023-08-03

**Soundness:** 4

**Excitement:**

3: Ambivalent: It has merits (e.g., it reports state-of-the-art results, the idea is nice), but there are key weaknesses (e.g., it describes incremental work), and it can significantly benefit from another round of revision. However, I won't object to accepting it if my co-reviewers champion it.

**Paper Topic And Main Contributions:**

The authors proposed MQA (Multi Query Attention) and GQA (Grouped-query attention) to overcome the memory bandwidth issue caused by the loading of multiple keys and values in the self-attention mechanism of the LM model.
Multi-query attention utilizes only a single key and value, resulting in faster computation speed but lower model capacity and quality. To address this limitation, the authors introduced GQA, which uses several grouped keys and values, achieving similar speed to MQA while maintaining comparable quality to the Multi-head approach.

**Questions For The Authors:**

1. What is the result of other versions such as 4, 16, 32 as well as GQA-8 in Table 1?

2. How does the performance of the model change when the alpha value exceeds 0.1?

3. Why did you apply MQA and GQA only to the decoder and not to the encoder?

4. What is the minimum number of groups to stabilize training with GQA?

**Reasons To Accept:**

1. Proposing a relatively simple yet effective methodology to reduce the computational overhead of LM.
2. Including experimental results with various parameter configurations for the suggested methodology.
3.  well-structured and easy to read paper.

**Reasons To Reject:**

1. lacking of detail about experiment design and settings.

1-a) Why  MQA and GQA are applied only to the decoder, while the encoder was left unaffected during the experiments.
1-b) Missing detail about the optimizer, the total number of epochs, and the time taken to train the entire dataset.

2. lacking of sufficient experimental results to support the claims made by the authors.

2-a) The authors mentioned that the model's training is unstable. Therefore, increasing the uptraining ratio (alpha) may lead to worse results. However, in Figure 5, it can be observed that the model's performance consistently improves as the maximum alpha ratio increases up to 0.1. There is a lack of experiment results regarding the point at which the performance starts to decline as the ratio further increases.

2-b) The authors only demonstrated in the paper that as the number of queries per group increases in GQA, the time also increases, as shown in Figure 6. However, it remains unclear how the performance actually changes with the increase in the number of queries per group.



3. The lack of detail about the claims.

3-a) GQA has superior training stability compared to MQA. However, there was no mention of the details of the correlation between group size and stability (how much group size guarantees stability) are not mentioned.

**Reproducibility:**

4: Could mostly reproduce the results, but there may be some variation because of sample variance or minor variations in their interpretation of the protocol or method.

**Reviewer Confidence:**

4: Quite sure. I tried to check the important points carefully. It's unlikely, though conceivable, that I missed something that should affect my ratings.

---

> ### Author Rebuttal · Authors · 2023-08-28
>
> We thank the reviewer for their extensive questions and comments.
>
> **Reason 1:**
>
> a. This is a question also raised by KbGZ, and reflects an important point about encoder-decoder inference, which we will clarify in the paper. The encoder representation can be computed in parallel, not token by token, and this computation is FLOPs bound, not memory bandwidth bound. Because MQA and GQA function by reducing memory bandwidth overhead, neither method is applicable to speeding up the encoder.
>
> b. Thanks, the mentioned details are indeed missing, and we will add them. The optimizer is Adafactor, with the same hyperparameters and inverse sqrt learning rate schedule as T5 for uptraining, and a constant 1e-3 learning rate for fine-tuning. The uptraining dataset is C4 for alpha% of 1T tokens, where alpha=5% took approximately 600 TPUv3 chip-days (see the original T5 paper for more details on the recipe and computational cost).
>
> **Reason 2:**
>
> a. We apologize for a point of confusion here. The instability is not in uptraining but rather the resulting fine-tuning. Moreover, it is not the case that increasing the uptraining ratio leads to worse results (in fact it leads to better results and more stable fine-tuning). Since the instability occurs primarily in the baseline method, and our GQA procedure solves the problem, we decided not to spend further resources on exploring the exact parameters where it occurs. We will clarify in the paper.
>
> b. This is a very valid point. Unfortunately, while profiling is cheap, fine-tuning large long-input models is very resource intensive, and we did not have the ability to evaluate performance for all values of the numbers of queries per group. We argue that GQA-8 is both very close to MQA in speed, and close to MHA in quality, and therefore must be close to optimal. Since then, the LLAMA2 work has also shown this to be a reasonable value.
>
>
> **Reason 3**:
> Due to limited computational resources, we did not experiment extensively with more than 8 query heads per KV head. In practice, more than 8 query heads per KV head does not lead to significant speedup and we do not expect this to be a common hyperparameter choice.
>
>
> **Question 1:** We direct the reviewer to our response to reason 2.b
>
> **Question 2:** Since improvement from alpha=0.05 to 0.1 is already modest, and since uptraining for longer is very costly, we did not explore higher values of alpha.
>
> **Question 3:** We direct the reviewer to our response to reason 1.a
>
> **Question 4:** We direct the reviewer to our response to reason 3.a

---

### Official Review · Reviewer_KbGZ · 2023-08-05

**Soundness:** 4

**Excitement:**

4: Strong: This paper deepens the understanding of some phenomenon or lowers the barriers to an existing research direction.

**Paper Topic And Main Contributions:**

The paper proposes uptraining for multi-query attention and transform multi-query attention to grouped-query attention where query heads are divided into  groups, each of which shares a single key head and value head. This leads to improvements in performance while maintaining the fast inference time.

**Questions For The Authors:**

1. How does applying GQA and MQA on the encoder with uptraining  affect performance?

**Reasons To Accept:**

1. The paper proposes that up-training and grouping of query-headed attention while maintaining fast inference of query-headed attention and similar performance of multi-headed attention. This is useful for time-constrained tasks.
2. The paper is easy to read,
3. The paper performs ablations on uptraining steps, number of groups

**Reasons To Reject:**

1. The authors uses time per sample to measure model efficiency. It may have been more useful to also report FLOPS to better compare with other methods.

**Reproducibility:**

3: Could reproduce the results with some difficulty. The settings of parameters are underspecified or subjectively determined; the training/evaluation data are not widely available.

**Reviewer Confidence:**

2: Willing to defend my evaluation, but it is fairly likely that I missed some details, didn't understand some central points, or can't be sure about the novelty of the work.

---

> ### Author Rebuttal · Authors · 2023-08-28
>
> We thank the reviewer for their questions and comments.
>
> **Reason 1:**
> Autoregressive decoding speed is typically bottlenecked by memory bandwidth (https://arxiv.org/abs/2211.05102) rather than FLOPs, as all model weights as well as the KV cache have to be loaded to SRAM for every token (see the original multi-query paper for a discussion). Multi-query and grouped-query attention increase speed not by decreasing the FLOPs, but rather by reducing the size of the KV cache and therefore memory bandwidth overhead.
>
> **Question 1:**
> This is a subtle point that we will clarify in the paper. The encoder representation can be computed in parallel, not token by token, and this computation is FLOPs bound, not memory bandwidth bound. Because MQA and GQA function by reducing memory bandwidth overhead, neither method is applicable to speeding up the encoder.

---

### Official Review · Reviewer_5bUv · 2023-08-05

**Soundness:** 2

**Excitement:**

3: Ambivalent: It has merits (e.g., it reports state-of-the-art results, the idea is nice), but there are key weaknesses (e.g., it describes incremental work), and it can significantly benefit from another round of revision. However, I won't object to accepting it if my co-reviewers champion it.

**Paper Topic And Main Contributions:**

This paper mainly studies how to improve the inference speed of large language models while maintaining the quality of the model. The author proposes an uptraining method to upgrade the existing multi-head model checkpoints to a multi-query model, and introduces a method called grouped-query attention (GQA) as a compromise between MQA and MHA . The experimental results show that the uptrained GQA is close to MHA in quality and close to MQA in operation speed.

**Reasons To Accept:**

1. The uptraining method proposed in this paper can effectively upgrade the existing MHA model to the MQA model, requiring only 5% of the computing power of the original pre-training, reducing the cost of model conversion.
2. The proposed GQA method achieves a high inference speed while maintaining the model quality, and provides an effective solution for improving the inference speed of large language models.
3. The experimental design is reasonable, and the results fully prove the effectiveness of the proposed method.

**Reasons To Reject:**

1. This article mainly focuses on the improvement of inference speed, but there is less discussion on the optimization of the model training phase.
2. Although the GQA method has achieved a good balance between quality and speed, there is still a certain decline in model capacity and quality. In the future, we can further explore how to improve model quality while maintaining speed.
3. If the restriction analysis points out that the evaluation method (Rouge score) is flawed, the current experiment may not fully reflect the performance of the model, and further exploration is necessary in this work.

**Reproducibility:**

4: Could mostly reproduce the results, but there may be some variation because of sample variance or minor variations in their interpretation of the protocol or method.

**Reviewer Confidence:**

2: Willing to defend my evaluation, but it is fairly likely that I missed some details, didn't understand some central points, or can't be sure about the novelty of the work.

---

> ### Author Rebuttal · Authors · 2023-08-28
>
> We thank the reviewer for their comments, and address their points below.
>
> **Reason 1:** We indeed focus on inference speed. Inference and training operate in different compute regimes; training is bottlenecked by FLOPs, while inference is bottlenecked by memory bandwidth (https://arxiv.org/abs/2211.05102). Multi-query and grouped-query attention improve inference speed by reducing memory bandwidth overhead, and therefore are not applicable for training. However, inference speed is a critical property for modern LLMs, since LLMs can receive billions of user requests per day while only being trained infrequently.
>
> **Reason 2:** We agree that there likely exist methods that achieve better trade-offs. Some examples of possible methods include pruning the least valuable KV heads, using more KV heads for important query heads, or conditionally loading some KV heads for only some tokens. However, we feel that GQA represents a significant step forward over MQA, and will independently be of value to the community (as recently seen in the LLAMA2 paper).
>
> **Reason 3:** Indeed further exploration of different evals would be valuable. We note that we evaluate tasks with a variety of metrics, including WMT (BLEU) and TriviaQA (F1). Moreover, the recent LLAMA2 work has demonstrated similar results with different eval methods, providing further evidence that our results are robust.

---

### Meta-Review · Area_Chair_eitA · 2023-09-07

**Recommendation:** 4

**Metareview:**

There is a consensus that this work is sound and exciting. The reviewers highlight that the method and paper are easy to understand and yield a straightforward inference improvement at negligible performance degradation. While some concerns have been raised regarding the evaluation in terms of ROUGE, the authors provide enough evidence of strong performance across other metrics.

---

### Decision · Program_Chairs · 2023-10-07

**Decision:**

Accept-Main

**Comment:**

There is a consensus that this work is sound and exciting. The reviewers highlight that the method and paper are easy to understand and yield a straightforward inference improvement at negligible performance degradation. While some concerns have been raised regarding the evaluation in terms of ROUGE, the authors provide enough evidence of strong performance across other metrics.